

# Epizootic ulcerative syndrome causes cutaneous dysbacteriosis in hybrid snakehead (*Channa maculata*♀ × *Channa argus*♂)

Zhifei Li[1,2], Guangjun Wang[1,2], Kai Zhang[1,2], Wangbao Gong[1,2], Ermeng Yu[1,2], Jingjing Tian[1,2], Jun Xie[1,2] and Deguang Yu[1,2]

[1] Key Laboratory of Tropical and Subtropical Fishery Resource Application and Cultivation, Pearl River Fisheries Research Institute, Chinese Academy of Fishery Sciences, Guangzhou, China
[2] Guangdong Ecological Remediation of Aquaculture Pollution Research Center, Guangzhou, China

## ABSTRACT

Cutaneous microbiota play an important role in protecting fish against pathogens. *Aphanomyces* infection causes epizootic ulcerative syndrome (EUS) in fish, and by perturbing the integrity of the cutaneous microbiota, increases the potential for infection by pathogenic bacteria. However, whether the composition of the cutaneous microbiota is altered in fish with EUS, and if so, which species are changed and how this might influence infected fish, is still largely unclear. Considering the importance of cutaneous microbiota in maintaining host health, we hypothesized that *Aphanomyces* infection significantly enhances the presence of certain bacterial pathogens in the cutaneous microbiota and causes cutaneous dysbacteriosis. To test this hypothesis, we compared the cutaneous microbiota compositions of hybrid snakehead (*Channa maculata*♀ × *Channa argus*♂) with and without *Aphanomyces* infection using Illumina Miseq sequencing of the 16S rRNA gene. Our results showed that the cutaneous microbiota of hybrid snakehead were significantly altered subsequent to EUS infection and that the numbers of potentially pathogenic bacteria classified into the genera *Anaerosinus*, *Anaerovorax*, *Dorea*, and *Clostridium* were significantly enhanced in the cutaneous microbiota of hybrid snakehead with EUS, whereas bacteria classified into the genera *Arthrobacter*, *Dysgonomonas*, *Anoxybacillus*, *Bacillus*, *Solibacillus*, *Carnobacterium*, *Lactococcus*, *Streptococcus*, *Achromobacter*, *Polynucleobacter*, *Vogesella*, and *Pseudomonas* were significantly reduced. These results imply that treatment for EUS should not only take into consideration the control of *Aphanomyces* reproduction but should also focus on regulating the cutaneous microbiota of infected fish.

## INTRODUCTION

Epizootic ulcerative syndrome (EUS) is a severely infectious disease of fish that is known to affect more than 100 types of cultured and wild fishes, and has caused serious losses in the

Corresponding authors
Jun Xie, xiejunhy01@126.com
Deguang Yu, gzyudeguang@163.com

aquaculture industry (*Lilley, Phillips & Tonguthai, 1992*; *Kamilya & Baruah, 2014*; *Gomo et al., 2016*; *Iberahim, Trusch & West, 2018*). The Organization International Des Epizooties (OIE) has confirmed that EUS is a seasonal epidemic caused by infection of the oomycete *Aphanomyces invadans* or *Aphanomyces piscicida*, and is listed among those diseases that must be notified (*OIE, 2013*; *Iberahim, Trusch & West, 2018*). *Aphanomyces* infection of fish occurs via water-borne secondary spores or planospores that adhere to the surface of impaired fish skin, a prerequisite for spore infection (*Lilley et al., 1998*; *Oidtmann, 2012*; *Kamilya & Baruah, 2014*), and subsequently develop into mycelium (*OIE, 2013*).

Fish skin primarily comprises an epidermis, mucus, and cutaneous microbiota, and represents an initial barrier in the defense against pathogens (*Gostin, Neagu & Vulpe, 2011*; *Boutin et al., 2014*; *Lazado & Caipang, 2014*). As a substantial component of the innate immune system of fish, the cutaneous microbiota play a vital role in the prevention of pathogenic invasion by forming a homeostatic microbial barrier (*Backhed, 2005*; *Stecher & Hardt, 2008*; *Loiseau et al., 2009*; *Harris et al., 2009*; *Boutin et al., 2014*). *Aphanomyces* infection causes the development of numerous red or gray ulcerous plaques in fish skin, resulting in the direct exposure of muscle to the surrounding water (*OIE, 2013*). This not only increases the risk of *Aphanomyces* infection but also increases the likelihood of invasion by bacterial pathogens (*Kamilya & Baruah, 2014*). A wide range of pathogenic bacteria have been cultured from the ulcerous plaques of fish with EUS. For example, *Lilley, Phillips & Tonguthai (1992)* isolated pathogenic bacteria belonging to four genera (*Aeromonas*, *Vibrio*, *Pseudomonas*, and *Micrococcus*), whereas *Thampuran et al. (1995)* reported that the compositions and relative abundances of the cutaneous microbiota of fish with EUS were as follows: *Aeromonas hydrophila* (40%), *Escherichia coli* (12.5%), *Arthrobacter* sp. (17.5%), *Acinetobacter calcoaceticus* (7.5%), *Enterobacter cloacae* (5%), *Pseudomonas aeruginosa* (5%), and *Micrococcus* sp. (5%). *Rahman et al. (2002)*, *Dhanaraj et al. (2008)*, and *Hossain (2009)* have also reported that bacteria in the genus *Aeromonas* predominate in the cutaneous microbiota of fish with EUS, and that these bacteria show strong pathogenicity. However, the identity of those bacterial species that are altered after EUS infection and their potential influence on hybrid snakehead (*Channa maculata*♀ × *Channa argus*♂) in situ remain unclear. The paucity of our current knowledge regarding changes in the cutaneous microbiota after EUS infection has precluded elucidation of the microecological mechanisms underlying EUS infection, and thereby limited the treatment of this disease. Considering the importance of cutaneous microbiota in defending fish against pathogens and the destructive effect of *Aphanomyces* infection on the cutaneous microbiota, we hypothesized that *Aphanomyces* infection significantly enhances the potential of infection by bacterial pathogens in the cutaneous microbiota of hybrid snakehead.

To test our hypothesis, we used Illumina MiSeq sequencing of the 16S rRNA gene to compare the cutaneous microbiota compositions of hybrid snakeheads with and without *Aphanomyces* infection, as well as the composition of microbiota from water and sediment samples collected from the pond inhabited by these fish. Our results provide novel insights that will contribute to the treatment of EUS.

## MATERIALS AND METHODS

### Sample collection and protocol

Specimens of hybrid snakehead with and without EUS infection were collected on April 12, 2016, from the pond of a hybrid snakehead aquaculture farm (113°07′53″E, 22°45′07″N; pond size 5,000 m$^2$ with a water depth of 1.8 m, water temperature 19.1 ± 2.9 °C, pH 6.60 ± 0.18) located in Junan Town, Foshan City, southern China. The hybrid snakeheads were naturally infected by *Aphanomyces* sp. and approximately 70% of individuals in the sampling pond were infected. Fish showing normal swimming and feeding behavior were also collected from the same pond. A total of 10 fish of each type were collected, and their body weight, body length, and area of ulceration were measured prior to collecting cutaneous microbiota samples. The areas of ulceration were measured using a two-scale grid-based method (*Zhou et al., 2016*; *Jiang et al., 2018*). For microbiota analysis, five fish of each type were sampled and analyzed. Sections of fish skin measuring 2 × 2 cm were collected from the ulceration caused by *Aphanomyces* infection, located on the right abdomen (near the fish tail) of the infected fish, under sterile conditions after washing three time using sterile distilled water. Skin samples were also collected from the same position in healthy control fish under sterile conditions. The skin samples were stored in five mL sterile centrifuge tubes at 4 °C and transferred to the laboratory for DNA extraction. We also collected two further fish skin samples from the areas of ulceration caused by *Aphanomyces* infection to identify fungal pathogens. In addition, five pond water and five sediment samples were collected from the same pond using a five-point sampling method for identification of the environmental microorganisms associated with the hybrid snakehead habitat. The pond water samples were collected at a depth of 50 cm using a one L water sampler and stored in one L sterile sampling bottles. The water temperature and pH were determined at each sampling site using a portable physicochemical analyzer (Hach, Loveland, CO, USA). Sediment samples were collected from the top zero to five cm layer of the sediment surface. All samples were transferred to the laboratory at 4 °C.

The experimental protocols used in this study were approved by the Animal Ethics Committee of the Guangdong Provincial Zoological Society, China, under permit number GSZ-AW002.

### Histopathological analysis

Three specimens of each type of hybrid snakehead with and without EUS were anesthetized using 80 mg/L MS-222, and from these fish, 2 × 2 cm samples of the epidermis and muscle (to a depth of approximately one cm) were collected from the right abdomen. These samples were thereafter cut into pieces of approximately 0.5 cm$^3$ and washed using cold (4 °C) deionized water to remove the mucus. Subsequently, the samples were fixed using 10% formalin solution and then stored in Davison's fixation solution. Paraffin tissue sectioning and hematoxylin-eosin staining were conducted according to previous reports (*Cardiff, Miller & Munn, 2014*; *Huang et al., 2018*). Photomicrographs of the stained sections were taken using a Nikon 80i microscopic imaging system.

## Pathogen identification

Two hybrid snakeheads with EUS were used to confirm the presence of *Aphanomyces*. A small piece from the area of ulceration of hybrid snakehead was used for DNA extraction according to a method described by *Fang et al. (2015)* and *Ni et al. (2017)*. A primer pair for the amplification of *Aphanomyces* DNA, F (5′-CTA GCC GAA GGT TTC GCA AGA-3′) and R (5′-GTT GTT TCC CAA TTT GCT TCC G-3′), was designed based on sequences of the *Aphanomyces* 18S rRNA gene in the GenBank database. The target fragment was 581 bp in length. Polymerase chain reactions (PCRs) were performed using 100 μL reaction mixes containing 1 × PCR buffer, one U of Taq DNA polymerase, 0.8 mM of each dNTP, 4.0 μM of each primer, and 40 ng microbial genomic DNA. The thermal cycling procedure consisted of an initial pre-denaturation step at 95 °C for 3 min, followed by 35 cycles of 95 °C for 30 s, 68 °C for 30 s, and 72 °C for 30 s, and a final extension at 72 °C for 5 min.

Genomic DNA of *Aphanomyces invadans* was used as positive control and sterile water was used as a negative control. Amplicons were checked using 1.5% agarose electrophoresis, and the length of target fragment was determined. The target fragment was retrieved from the gel using a gel extraction kit (Omega, Doraville, GA, USA) and was subsequently sequenced using an ABI 3700 DNA sequencer (ABI, Foster City, CA, USA).

## Cutaneous microbiota enumeration

Five fish of each type were sampled and used to enumerate the cutaneous microbiota. The cutaneous microbiome was eluted from fish skin by placing skin samples into two mL of sterile PBST (140 mM NaCl, 10 mM phosphate, three mM KCl, 0.05% Tween-20, pH 7.4) in a sterile five mL conical tube, and mixing by vortexing for 2 min. Bacteria were collected from the PBST suspension by centrifugation at $5,000 \times g$ for 5 min. The DNA was extracted and purified using a genomic DNA clean & concentrator kit (Zymo, Irvine, CA, USA).

Polymerase chain reactions amplification was carried out using the prokaryotic primer set 515F (5′-GTG CCA GCM GCC GCG GTA A-3′) and 806R (5′-GGA CTA CHV GGG TWT CTA AT-3′) for the V4 region of the 16S rRNA gene, as we have described previously (*Tamaki et al., 2011*; *Li et al., 2016*, *2017*; *Ni et al., 2018*). The resulting PCR products were pooled and purified using an Axygen gel extraction kit (Axygen, Union City, CA, USA). After purification, the PCR products of the 16S rRNA V4 region were quantified using a NanoDrop spectrophotometer (Thermo Fisher Scientific, Waltham, MA, USA). A mixture of the amplicons was then used for sequencing using the Illumina MiSeq platform at Beijing Novogene Technology Co., Ltd, Beijing, China.

The sequencing data were analyzed as described previously (*Ni et al., 2017*; *Huang et al., 2018*; *Li et al., 2018*) with appropriate modifications. Briefly, after sequencing, the paired-end reads were overlapped to assemble the V4 tag sequences using Flash software (*Magoč & Salzberg, 2011*). The primers and spacers were trimmed using QIIME 1.9.0 (*Caporaso et al., 2010*). To minimize the effects of random sequencing error, both the low-quality fragments and sequences shorter than 240 bp were removed. Prior to further analysis, we checked for the presence of chimeras, which were filtered out using UCHIME software (*Edgar et al., 2011*). The sequences were classified into operational taxonomic units (OTU) by setting a 0.03

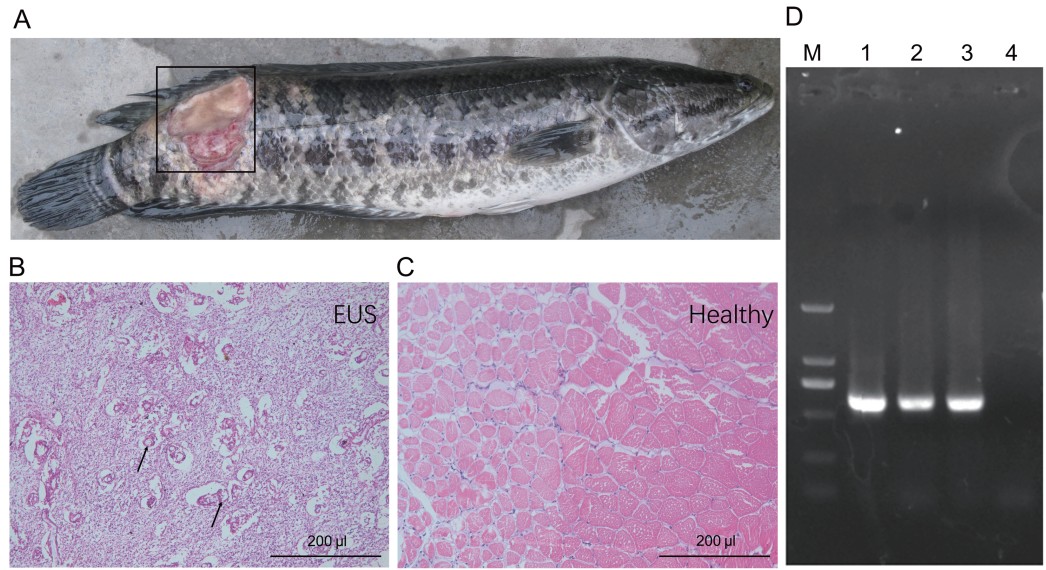

**Figure 1 Gross lesions (A) and hematoxylin and eosin-stained paraffin tissue sections of hybrid snakehead (*Channa maculata*♀ × *Channa argus*♂) without (B) and with (C) epizootic ulcerative syndrome (EUS), and an electrophoresis profile of (i) Aphanomyces (D).** M, Marker2000; (1) positive control using the genomic DNA of *A. invadans*; (2) and (3) the amplification of samples collected from an ulcerated region of hybrid snakehead with EUS; (4) negative control using sterile water. The black frame in panel (A) indicates the ulcerated region of hybrid snakehead with EUS. Black arrows in panel (B) indicate the infection foci of hybrid snakehead with EUS.               

distance limit using the CD-HIT program (*Li & Godzik, 2006*). Taxonomic assignments of each OTU were determined using the RDP classifier (*Wang et al., 2007*).

Weighed and unweighted UniFrac distances between samples were calculated and principal co-ordinates analysis was conducted using QIIME 1.9.0. Principal component analysis and non-parametric multivariate analysis of variance (PERMANOVA) (*Anderson, 2001*) were used to reveal differences in the microbiota of the different groups and were conducted using the vegan package of R 3.5.1 (*Dixon, 2003*). A non-parametric Kruskal–Wallis test was used to detect significant differences among the different groups, and box plots were drawn to depict the relative abundances of significantly different OTUs among groups using STAMP software (*Parks et al., 2014*). Statistically significant markers were added to the box plots using Adobe Illustrator CS5 software according to the post hoc test results. A network of samples based on the dominant genus relative co-abundance between samples was constructed using the igraph and psych packages of R 3.5.1. Linear discriminant analysis effect size (LEfSe) was conducted to screen the significantly different genera between EUS and healthy controls using the Galaxy platform (*Segata et al., 2011*).

Sequencing data obtained in this study have been submitted to the NCBI Sequence Read Archive database with the accession number PRJNA495472.

## RESULTS

### Skin damage and the presence of *Aphanomyces*

Muscle underlying the damaged skin of hybrid snakehead with EUS became ulcerated and were colonised by numerous filamentous fungi (Figs. 1A and 1B). Control muscle
**Table 1 Characteristics of the hybrid snakehead (*Channa maculata*♀ × *Channa argus*♂) examined in the present study (*n* = 10).**

| Parameter | EUS group | Healthy control | *p*-value |
|---|---|---|---|
| Sample size (N) | 10 | 10 | |
| Body weight | 658.5 ± 111.7 | 716.2 ± 213.6 | NS |
| Body length | 39.1 ± 3.1 | 40.7 ± 5.9 | NS |
| Ratio of ulcer area to body surface area (%) | 12.2 ± 3.9 | 0 | |
| OD$_{630nm}$ of skin suspensions | 0.09 ± 0.03 | 0.05 ± 0.02 | 0.046 |

Notes:
An independent *t*-test was used to detect significant differences between the epizootic ulcerative syndrome (EUS) and Healthy control groups. Quantitative data are presented as the mean ± standard deviation (SD).
NS, no significant difference.

shown for comparison (Fig. 1C). Numerous mycotic granulomas and large amounts of mycelia, which are typical characteristics of EUS, were detected in the ulcerated muscle of hybrid snakehead with EUS (Fig 1B). Ulceration in hybrid snakehead with EUS covered 12.2% ± 3.9% of skin area (Table 1). Infected fish were observed to be sluggish and lacking in vitality. Moreover, we found that the OD$_{630nm}$ of skin suspensions obtained from fish with EUS was significantly higher than that of suspensions from healthy controls (independent *t*-test, $t = 2.44$, $p = 0.046$), indicating that a greater number of bacteria probably parasitize the skin of fish with EUS. PCR results showed that target DNA fragments (approximately 580 bp) could be amplified from the samples collected from the ulcerated regions of hybrid snakehead with UES (Fig. 1D). Sequencing results for the amplified DNA fragments showed that the fragments were derived from *Aphanomyces* species, indicating that *Aphanomyces* was the causal pathogen of EUS in the hybrid snakehead we analyzed.

## Changes in the cutaneous microbiota of hybrid snakehead with EUS

Following the removal of low-quality reads and chimeric sequences, we obtained a total of 999,539 sequences from the 20 samples used for analysis (five cutaneous microbiota samples from hybrid snakehead with EUS, five cutaneous microbiota samples from healthy hybrid snakehead, five pond water microbiota samples, and five pond sediment microbiota samples). To eliminate the influence of sequencing depth, 29,781 sequences of each sample were randomly re-sampled according to the amount of sample sequence to obtain the minimum effective sequences and used for further analysis. Although three archaeal and 56 bacterial phyla were detected from the 20 microbiota samples, we found that the microbiota primarily comprised 15 predominant phyla (their relative abundances were greater than 1% in at least one sample; Fig. 2A). Although the relative abundances of most of these dominant phyla differed significantly among the cutaneous microbiota of hybrid snakehead with EUS, and those of healthy hybrid snakehead, pond water microbiota, and sediment microbiota, only Actinobacteria and Firmicutes were significantly reduced in the cutaneous microbiota of hybrid snakehead with EUS compared with that of healthy hybrid snakehead (Fig. 2C, 2I). This could be attributable to the fact that the numbers of many unclassified bacteria were enhanced in the cutaneous microbiota of hybrid snakehead with EUS, representing up to 29.77% ± 14.92% of the analyzed

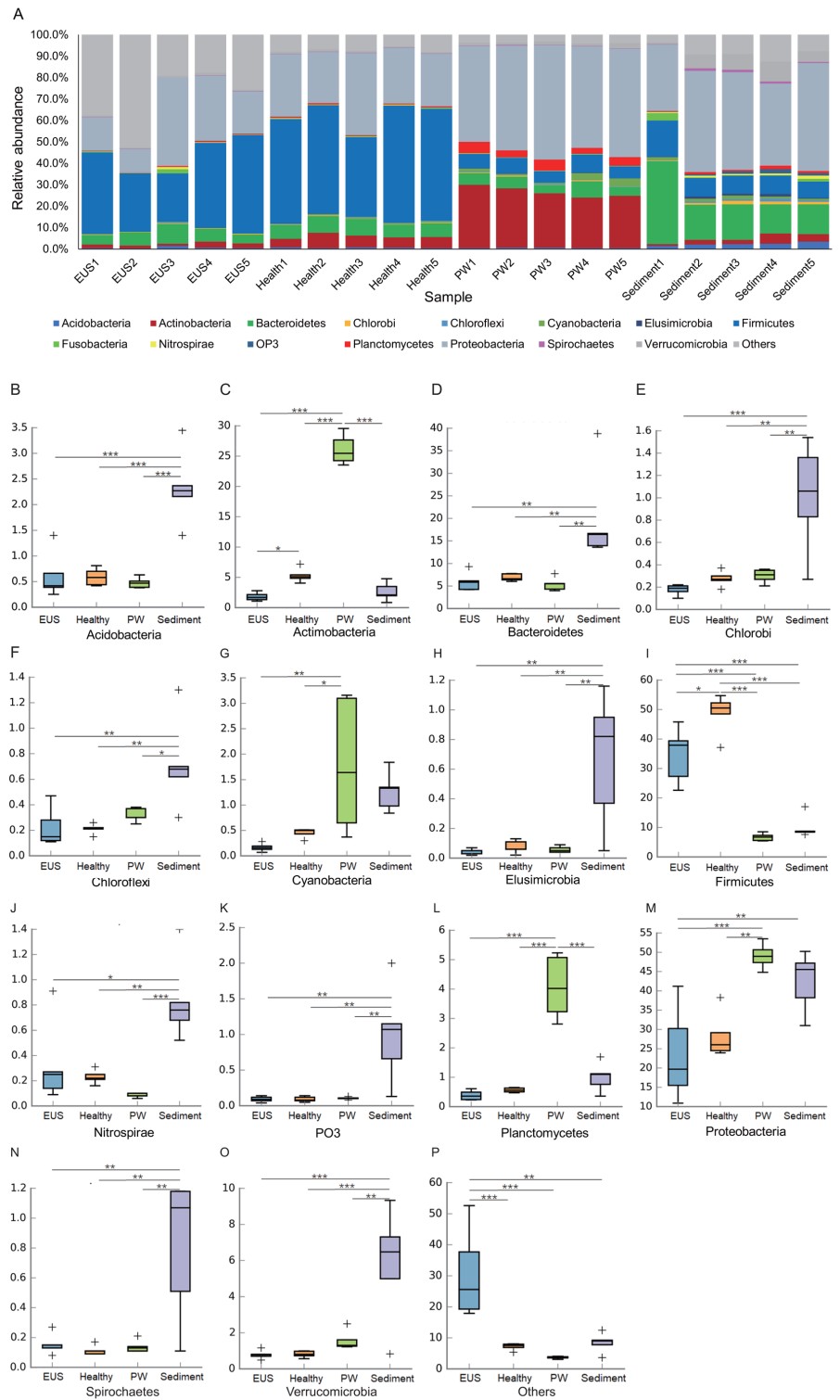

**Figure 2 Dominant phyla of the cutaneous microbiota of hybrid snakehead (A) and box plots of significantly different dominant phyla (B–P).** EUS, cutaneous microbiota of hybrid snakehead with epizootic ulcerative syndrome; Healthy, cutaneous microbiota of healthy hybrid snakehead; PW, pond water microbiota; S, sediment microbiota. The non-parametric Kruskal–Wallis test was used to detect significant differences among different groups. $^*p < 0.05$, $^{**}p < 0.001$, $^{***}p < 0.0001$.

sequences. In contrast, unclassified bacteria comprised only 6.33% ± 1.07% of sequences in the cutaneous microbiota of healthy fish (independent $t$-test, $t = 5.303$, $p = 0.024$).

In the present study, we detected a total of 7,108 OTUs classified into 969 genera, among which, 201 genera dominated the cutaneous microbiota and habitat microbiota. Streptococcaceae unidentified genus (24.48% ± 5.37%), genus *Lactococcus* (7.97 ± 1.46%), genus *Clostridium* (3.79% ± 0.37%), Comamonadaceae unidentified genus (3.70% ± 0.30%), and Betaproteobacteria unidentified genus (2.48% ± 1.19%) were the five most abundant genera in the cutaneous microbiota of healthy hybrid snakehead, whereas genus *Clostridium* (16.37% ± 6.71%), Peptostreptococcaceae unidentified genus (6.24% ± 7.25%), Comamonadaceae unidentified genus (2.73% ± 1.98%), JTB215 unidentified genus (2.69% ± 1.27%), and *Betaproteobacteria* unidentified genus (2.48 ± 2.52%) were the five most abundant genera in the cutaneous microbiota of hybrid snakehead with EUS. At both the OTU (PERMANOVA, $F = 12.746$, $p < 0.001$; Fig. 3A) and genus (PERMANOVA, $F = 16.409$, $p < 0.001$; Fig. 3B) levels, there were significant difference among the microbiota from the cutaneous microbiota of hybrid snakehead with EUS, cutaneous microbiota of healthy controls, pond water, and pond sediment (Figs. S1A and S1B). The unweighted UniFrac distances in the EUS group were significantly higher than those in the PW group (Wilcoxon test, $p = 0.001$; Fig. S1C; Table S2). The weighted UniFrac distances in the EUS group were significantly higher than those in the healthy group (Wilcoxon test, $p < 0.001$) and the PW group (Wilcoxon test, $p = 0.001$; Fig. S1D; Table S2). The weighted UniFrac distances between groups were significantly higher than those within groups (Fig. S1D; Table S2). The LEfSe results indicated that bacteria classified into the genera *Anaerovorax*, *Anaerosinus*, *Dorea*, and *Clostridium* were significantly enhanced in the cutaneous microbiota of hybrid snakehead with EUS, whereas those classified into the genera *Arthrobacter*, *Dysgonomonas*, *Anoxybacillus*, *Bacillus*, *Solibacillus*, *Carnobacterium*, *Lactococcus*, *Streptococcus*, *Achromobacter*, *Polynucleobacter*, *Vogesella*, and *Pseudomonas* were significantly reduced (Fig. 3C).

The dominant OTUs enhanced in the cutaneous microbiota of hybrid snakehead with EUS tended to be positively correlated with the dominant OTUs in the sediment, whereas dominant OTUs in the cutaneous microbiota of healthy hybrid snakehead tended to be positively correlated with those reduced in the cutaneous microbiota of hybrid snakehead with EUS or those in the sediment (Fig. 4). The dominant OTUs reduced in the cutaneous microbiota of hybrid snakehead with EUS tended to positively correlated with those in the pond water (Fig. 4). These results indicate that the dominant cutaneous microbiotas of hybrid snakehead with EUS were more closely related to the sediment microbiotas, whereas those of healthy hybrid snakehead were closer to the pond water microbiotas.

## DISCUSSION

Cutaneous microbiota play a vital role in the maintenance of health and prevention of pathogenic invasion by forming a homeostatic microbial barrier (*Backhed, 2005*; *Stecher & Hardt, 2008*; *Loiseau et al., 2009*; *Harris et al., 2009*; *Boutin et al., 2014*; *Federici et al., 2015*). Our results indicate that *Aphanomyces* infection significantly perturbs the

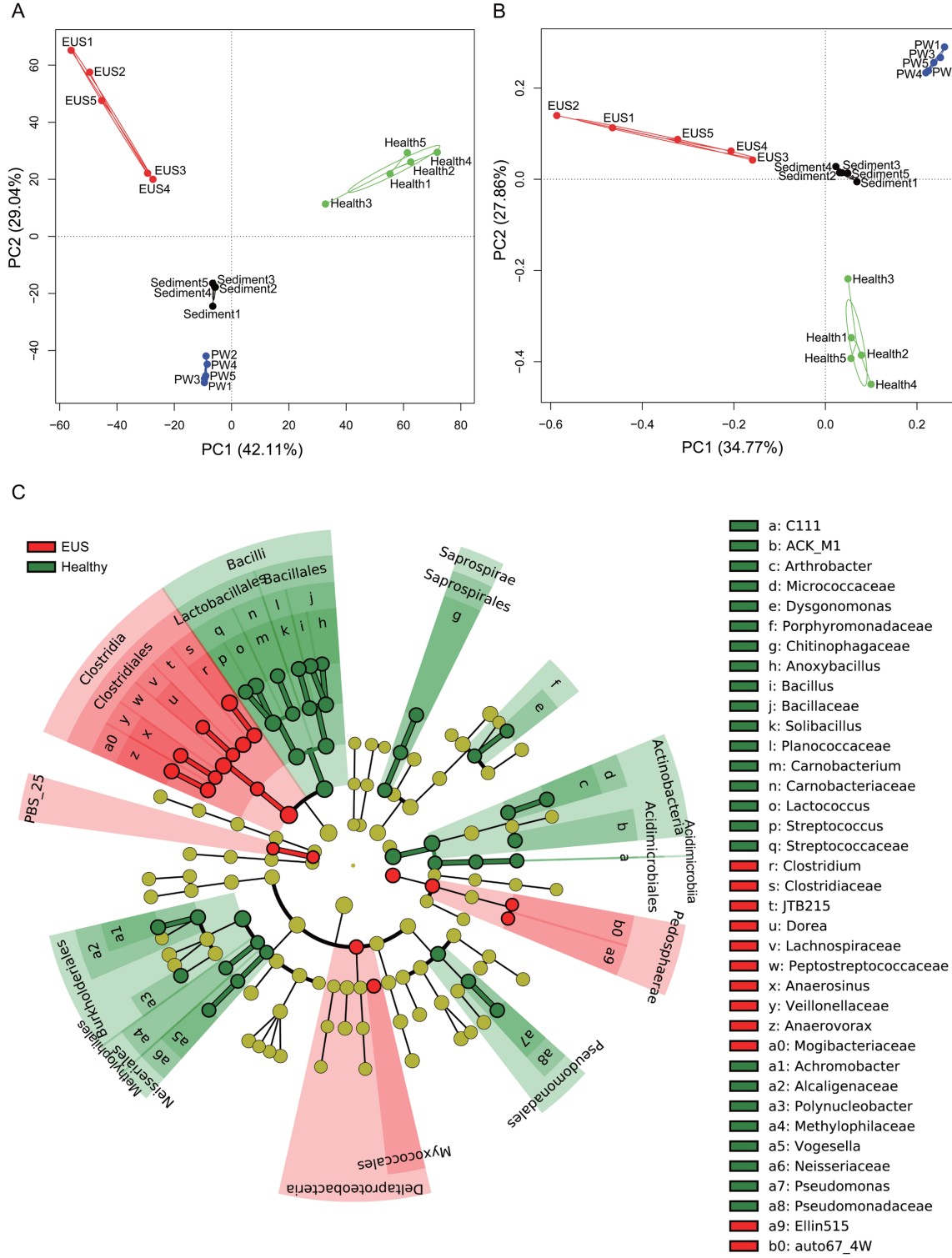

**Figure 3 Principal component analysis (PCA) profiles of microbiota at the OTU (A) and genus (B) levels, and LEfSe profile (C).** LEfSe profile (C) showing the significant differences of dominant genera in the cutaneous microbiota between hybrid snakehead with and without epizootic ulcerative syndrome (EUS).

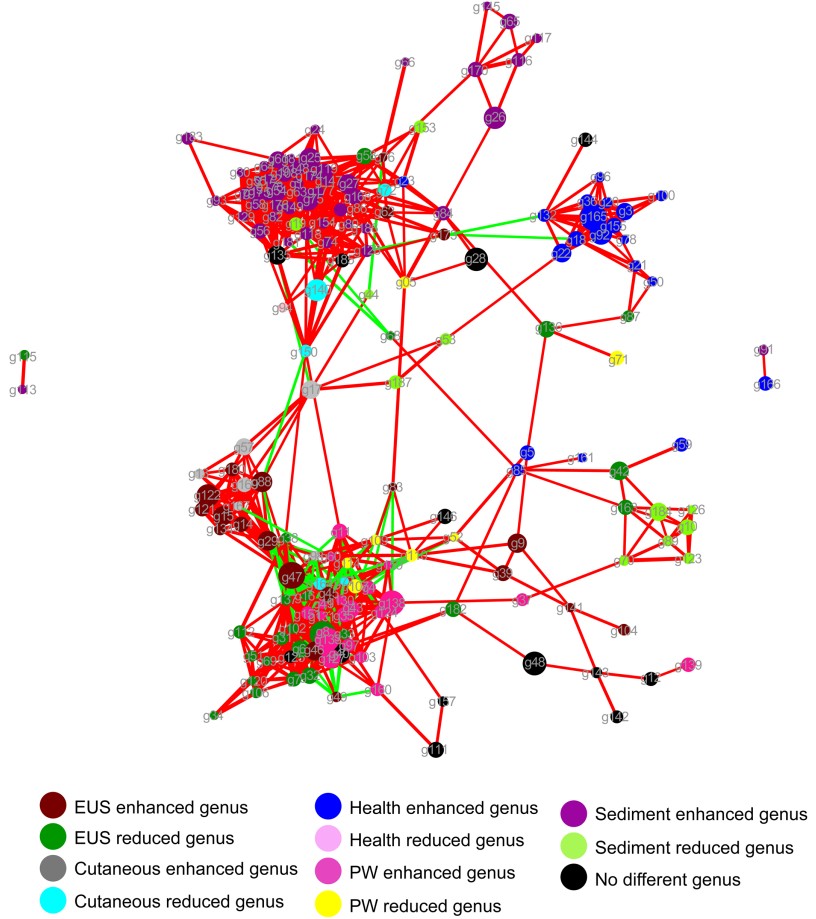

**Figure 4 Network showing the co-occurrences of dominant bacterial genera in the cutaneous microbiota of hybrid snakehead and the pond water and sediments of their habitat.** Each dot (node) represents a sample. A link (line) between two dots highlights a Spearman correlation index > 0.7 between the two dots and a Bonferroni-corrected *p*-value < 0.05. Red lines denote a positive correlation between two dots, whereas green lines denote a negative correlation between two dots. The thickness of the lines signifies the value of the Spearman correlation index. The size of nodes indicates the totals of the relative abundances of genera in all samples. PW, pond water microbiota. The genus IDs corresponding to phylogenetic genera are shown in Table S1.

composition of the cutaneous microbiota in hybrid snakehead (Fig. 3). Although *Dhanaraj et al. (2008)* reported that *Aeromonas hydrophila*, *Enterobacter* sp., *Vibrio* sp., *Pseudomonas* sp., *Escherichia coli*, *Aphanomyces invadans*, and *Aspergillus* sp. were the predominant isolated and identified species in EUS-infected murrel (*Channa striatus*) in Tirunelveli, India, we found that genus *Clostridium* (16.37% ± 6.71%), Peptostreptococcaceae unidentified genus (6.24% ± 7.25%), Comamonadaceae unidentified genus (2.73% ± 1.98%), JTB215 unidentified genus (2.69% ± 1.27%), and Betaproteobacteria unidentified genus (2.48% ± 2.52%) were the five most abundant genera in the cutaneous microbiota of hybrid snakehead with EUS. Differences in habitat, host fish, and analytical methods may explain the disparate results obtained in different studies; however, given that many bacteria cannot be cultured (*Turroni et al., 2008*), we believe that our results are probably more accurate.

The skin mucus layer is also a medium that potentially plays an antibacterial role (*Esteban, 2012*), which may be dependent on multiple factors, including different environmental conditions and intra- or interspecific variation (*Fast et al., 2002*; *Nigam et al., 2012*). The mucus layer is inhabited by a multitude of microorganisms and it is constantly exposed to many others present in the aquatic environment (*Tapia-Paniagua et al., 2018*). In the present study, our results showed that subsequent to *Aphanomyces* infection, bacteria classified into the genera *Anaerovorax*, *Anaerosinus*, *Dorea*, and *Clostridium* were significantly enhanced in the cutaneous microbiota of hybrid snakehead, whereas those classified into the genera *Arthrobacter*, *Dysgonomonas*, *Anoxybacillus*, *Bacillus*, *Solibacillus*, *Carnobacterium*, *Lactococcus*, *Streptococcus*, *Achromobacter*, *Polynucleobacter*, *Vogesella*, and *Pseudomonas* were significantly reduced. Most of the enhanced genera, particularly *Anaerovorax*, *Dorea*, and *Clostridium*, are commonly reported components of mammalian gut microbiota and their numbers tend to be correlated with host metabolic disturbance or infection (*Yildirim et al., 2010*; *Lahti et al., 2013*; *Buffie & Pamer, 2013*; *Ng et al., 2013*; *Bhattarai, Pedrogo & Kashyap, 2017*). However, their potential influence on host health in hybrid snakehead is still unclear. Many of the genera with reduced abundance in the cutaneous microbiota of hybrid snakehead with EUS, including *Streptococcus*, *Carnobacterium*, and *Bacillus*, contain species that are widely reported as probiotics (*Salinas et al., 2006*; *Lazado, Caipang & Estante, 2015*), although, conversely, some bacteria in the genus *Streptococcus* are also reported as pathogens (*Eldar, Bejerano & Bercovier, 1994*; *Bachrach et al., 2001*). However, little is currently known regarding the dynamics of these groups in the cutaneous microbiota.

Despite the fact that EUS is a frequently occurring disease and causes serious losses in the aquaculture industry (*Catap & Munday, 2002*; *Gomo et al., 2016*; *Iberahim, Trusch & West, 2018*), to date no effective and practical treatment approach has been proposed for this disease. Although water temperature is known to influence the reproduction of *Aphanomyces* (*Chinabut et al., 1995*), regulating water temperature is not a viable option in outdoor ponds. Diets supplemented with an ethanolic extract of *Rauvolfia tetraphylla* leaves have been reported to enhance innate immunity and confer resistance to *Aphanomyces invadans* infection in Indian major carp *Labeorohita* (*Yogeshwari et al., 2015*), whereas *Shanthi Mari et al. (2014)* have reported modulation of the immune system of infected *Cirrhina mrigala* fish fed with a 1% chitin- or chitosan-enriched diet, which thereby confers disease resistance against *Aphanomyces invadans*. Although the results of the present study indicate that *Aphanomyces* infection significantly perturbs the cutaneous microbiota of hybrid snakehead, the treatment of EUS should not only consider the control *Aphanomyces* reproduction but also aim to regulate the cutaneous microbiota of infected fish.

Habitat and seasonality are the major factors influencing the cutaneous microbiota of fish (*Boutin et al., 2013*; *Larsen et al., 2015*). In addition, the cutaneous microbiota differ significantly from the microbiota in the surrounding aquatic environment, as shown in the present study (Figs. 3A and 3B; Figs. S1A and S1B). For instance, *Larsen et al. (2015)* have reported that the cutaneous microbiota of *Fundulus grandis* is significantly different from that in its marine habitat. The cutaneous microbiota of *F. grandis* is

dominated by *Gammaproteobacteria* and *Betaproteobacteria*, whereas the marine microbiota is dominated by *Cyanobacteria* and *Alphaproteobacteria*. The status of fish, for example whether wild or farmed, has also been reported to significantly influence the cutaneous microbiota. *Boutin et al. (2013)* reported that the cutaneous microbiota of *Salvelinus fontinalis* is dominated by Proteobacteria (70.6%), followed by Actinobacteria (26.4%) and Bacteroidetes (2.9%). However, *Minniti et al. (2017)* reported that the cutaneous microbiota of farmed *Salmo salar* is dominated by Proteobacteria, followed by Firmicutes and Acidobacteria. In the present study, our results indicated that Firmicutes, Proteobacteria, and Bacteroidetes dominate the cutaneous microbiota of farmed hybrid snakehead. These finding imply that colonization by Firmicutes is probably enhanced in an aquaculture environment.

Skin lesions are among the most important infection pathways in fish and the cutaneous microbiota play a vital role in prevention of pathogenic invasion. However, compared with gut microbiota, the composition and function of the bacteria comprising the cutaneous microbiota are still rarely studied in fish. As a potential basis for the prevention and treatment of cutaneous infected fish, the cutaneous microbiota of fishes need to be more thoroughly analyzed in the future. In addition, studying the composition of microbial communities inhabiting skin ulcers will contribute to enhancing our understanding of ulcer etiology (*Karlsen et al., 2017*).

## CONCLUSIONS

*Aphanomyces* infection significantly perturbs the cutaneous microbiota of hybrid snakehead. In the present study, we found that potentially pathogenic bacteria classified into the genera *Anaerovorax*, *Anaerosinus*, *Dorea*, and *Clostridium* were significantly enhanced in the cutaneous microbiota of hybrid snakehead infected by the oomycete *Aphanomyces*, whereas those classified into the genera *Arthrobacter*, *Dysgonomonas*, *Anoxybacillus*, *Bacillus*, *Solibacillus*, *Carnobacterium*, *Lactococcus*, *Streptococcus*, *Achromobacter*, *Polynucleobacter*, *Vogesella*, and *Pseudomonas* were significantly reduced. These results imply that treatment for EUS should not only take into consideration the control of *Aphanomyces* reproduction but should also focus on regulating the cutaneous microbiota of infected fish. These findings expand our current knowledge regarding the relationship between cutaneous bacterial microbiota and EUS.

## ACKNOWLEDGEMENTS

We thank Jiajia Ni at Guangdong Meilikang Bio-Science Ltd., China, for assistance with the data statistics and analysis.

### Funding

This work was supported by the National Key Technology R&D Program of China (Project No. 2012BAD25B04; 2012BAD25B01). The funders had no role in study design, data collection and analysis, decision to publish, or preparation of the manuscript.

## Grant Disclosure

The following grant information was disclosed by the authors:

The National Key Technology R&D Program of China: 2012BAD25B04; 2012BAD25B01.

## Competing Interests

The authors declare that they have no competing interests.

## Author Contributions

- Zhifei Li conceived and designed the experiments, performed the experiments, analyzed the data, contributed reagents/materials/analysis tools, prepared figures and/or tables, authored or reviewed drafts of the paper, approved the final draft.
- Guangjun Wang conceived and designed the experiments, performed the experiments, analyzed the data, contributed reagents/materials/analysis tools, prepared figures and/or tables, authored or reviewed drafts of the paper.
- Kai Zhang analyzed the data, contributed reagents/materials/analysis tools.
- Wangbao Gong conceived and designed the experiments, performed the experiments, analyzed the data, authored or reviewed drafts of the paper.
- Ermeng Yu analyzed the data.
- Jingjing Tian conceived and designed the experiments, analyzed the data.
- Jun Xie conceived and designed the experiments, authored or reviewed drafts of the paper, approved the final draft.
- Deguang Yu conceived and designed the experiments, authored or reviewed drafts of the paper, approved the final draft.

## Animal Ethics

The following information was supplied relating to ethical approvals (i.e., approving body and any reference numbers):

The Animal Ethics Committee of the Guangdong Provincial Zoological Society, China approved this research (permit number GSZ-AW002).

## Data Availability

Sequencing data are available at the NCBI Sequence Read Archive database, accession number PRJNA495472.

## Supplemental Information

Supplemental information for this article can be found online at http://dx.doi.org/10.7717/peerj.6674#supplemental-information.

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
