# Peer review of "Epizootic ulcerative syndrome causes cutaneous dysbacteriosis in hybrid snakehead (Channa maculata♀ × Channa argus♂)"

_PeerJ, doi:10.7717/peerj.6674_

## Round 0.1 · original submission · Major Revisions

I support all comments of the two reviewers. I agree that it is really great to see histology. However, please make sure that all the pathological changes described in the text are visible in the figures, for example the figures are at too low magnification to show all the detail mentioned in the text (I suggest including high magnification inset or another figure), I cannot see skin in figure of the EUS affected fish which is cited in the text to support statements about skin changes. Please use arrows or letters to show features of interest in the figures. Please be sure to make all recommended changes, including improving histology images and address all the comments raised by the reviewers.

·

Basic reporting

Clear and unambiguous, professional English used throughout.
--> The primary manuscript yes, the animal protocols no
--> Animal protocols and approval are not in English, I’m not familiar with PeerJ’s official regulations, but it would be good to have it translated.

Literature references, sufficient field background/context provided.
--> The problem of EUS is well documented and defined in the introduction. Understanding secondary infections and disease susceptibility in a host is a very important part of medical microbiology. Further, few research studies have been conducted looking at microbial changes in a host while infected with a eukaryotic parasite.

Professional article structure, figs, tables. Raw data shared.
--> Raw data has not been made publically available

Experimental design

Original primary research within Aims and Scope of the journal
--> Yes

Research question well defined, relevant & meaningful. It is stated how research fills an identified knowledge gap.
--> Yes

Rigorous investigation performed to a high technical & ethical standard.
Methods described with sufficient detail & information to replicate.
--> No, see below
#Sampling methodology Line 86-106
It is not clear how infected fish were obtained. Were fish inoculated with an infectious dose and if so how much? Were fish just picked from the pond and separated into healthy vs sick? Were there various levels of infection rate among the fish? Line 90-91 states “
90 Five fish of each type were collected, and their body weight, body length, and area of ulceration were measured prior to collecting cutaneous microbiota samples.”
Line 94: what is the justification of washing off the mucus prior to sequencing… many of the protective microbes likely will be removed by this procedure
Where are the results for fungal and bacterial isolates… Line 98-fungal pathogen ID, Line 97-bacteria culturing

#Histopathological analysis, Line 107-116
It is fantastic that the authors are doing histopathology analysis on their samples, but there is no mention as to what they have actually done (measured). Also, there is no mention of statistics used to indicate disease state

#Pathogen ID Line 117-133
does the author describe rates of disease ?
#Microbiome analysis Line 133 – 164
How many samples were actually sequenced…and were there appropriate negative and positive controls included? Line 134-138 does not mentioned sample numbers
There are many other citations to using this 16S rRNA region, you should cite the other original papers rather than just your labs, Line 141
Using 97% OTU picking is no longer the best method for doing OTU picking as it has been replaced with ‘exact-sequence-variant’ procedures. Examples include dada-2 and deblur. Line 155
Line 157-158, the MANOVA test can only be used on normal-distributed data, it is NOT non-parametric. You may use the Kruskal-wallis test as a means to compare categorical data and is essentially the ‘non-parametric anova’ test. The PERMANOVA, which is what you reference in that citation is a non-parametric statistic which can be used on non-normal data. Also, it is not appropriate to pull out individual taxa and compare against other taxa from other groups (figure 2b). There are a variety of methods out there which mostly involve log-ratio comparisons

Methods described in with sufficient detail
--> for the most part yes, although there are some discrepancies which need to be addressed and are described in previous section

Validity of the findings

Impact and novelty not assessed. Negative/inconclusive results accepted. Meaningful replication encouraged where rationale & benefit to literature is clearly stated.
Data is robust, statistically sound, & controlled.

--> Some of the statistical measures used are no longer acceptable to microbiome research and thus should be recalculated using appropriate methods. For instance, due to the compositional nature of microbiome data, it is almost always non-parametric, thus using parametric statistics is not appropriate.
--> Line 171-172, is there a way to better describe (descriptive statistics) the level and types of ulceration statistically?
-->Line 176-177, To quantify bacterial growth on the skin you compare OD readings, which is fine. You could in addition compare the 16S copy numbers or a single copy gene of the actual samples to quantify overall microbial density amongst the two groups (healthy vs disease)
-->Line 181, Did you process samples from both health and diseased fish and show relative concentrations of the Aphanomyces amplicon across these sample groups? Figure 1C only shows that you tested presumably 4 samples, whereas you describe sampling 20 for comparisons in Table 1 and 10 for the microbiome. Please provide all of the data including if Aphanomyces was at all detected on the healthy fish. Alternatively, do qPCR to show relative concentrations in the two groups.
-->Line 187, How did you determine your rarefraction depth – this seems random. Did you include controls and if so what were their read counts?
-->Line 200-226, please repeat the analysis using appropriate statistical measures as commented on in the methods section. Note, the LEfSe is appropriate but you do not mention it in your methods.
-->Line 212, You use the LEfSe method to do differential abundances but you don’t describe this in your methods section…
-->Line 224-226, This is a very interesting finding/observation, but you do not actually test this with statistics. You should calculate the actual beta-diversity distances of each grouping (healthy vs sediment, healthy vs water, disease vs sediment… etc) and then compare the overall similarities to test your observation. Currently, the network analysis does not do this statistically. Your PCA plots in figure 3 provide some support for this, but again is not tested. Also, it would be better to use weighted and unweighted unifrac distances which takes into account the phylogenetic distances of microbes in samples followed by a PCoA visualization. There are many inherent problems with visualizing microbiome data when comparisons are made using Euclidean distances.

Conclusion are well stated, linked to original research question & limited to supporting results.

--> The authors conclude that upon Aphanomyces infection the cutaneous microbiome is perturbed and that furthermore other (presumably secondary) pathogens are enriched which could further cause harm to the fish. The author identifies microbes which are also reduced in this given scenario. The authors do not perform any additional experiments to evaluate and validate that the ‘enriched pathogens’ would actually be pathogenic by Koch’s postulates. They also do not describe any functional aspects such as toxin genes to validate whether these microbes are indeed pathogenic. Since this is the main argument of the study, it would be important to validate. This could be done by whole genome sequencing or targeted PCR of those toxin genes. Further, pathogens could be isolated and cultured and then inoculated onto fish to confirm pathogenesis.
There have been several other microbiome papers associated with this aquaculture important fish species. It would have been nice to see a meta-analysis to understand how these microbes which were abundant fit into the broader context.

Reviewer 2 ·

Basic reporting

The manuscript was well written and with very concise presentation of experimental results. The authors provided adequate background or overview of the study with references that are relevant to EUS and bacterial infections associated with the syndrome. The authors clearly stated the hypothesis and the objectives of the study.

Some minor corrections:
list of references:
lines 315-316 - revise the title of the article to be consistent with the format.
line 306 - scientific name should be italicised.
line 402 - ulcerativesyndrome should be ulcerative syndrome

Experimental design

1. The authors stated in the "Materials and Methods" section that 5 fish each (with EUS and without EUS) were collected, but in Table 1, the sample size for each group was indicated as "10 fish". Perhaps the authors can check this or clarify how they used the 10 fish that were collected (5 each for histopathology and 5 each for microbiome analyses??).

2. The authors collected both hybrid snakehead with EUS and without EUS from one pond, kindly indicate the size of the pond where the fish were collected. Although the fish were collected on the same day, the environmental factors (eg. water temperature, pH) were reported with standard deviations (+/-). The authors need to clearly state if this was daily variations or if this indicates fluctuations within a specific period of time (eg, week, month).

Validity of the findings

1. The conclusions were clearly stated and answered the hypothesis posed by the authors. However, the study also presented bacterial composition of pond water and sediments. These data were not addressed in the discussion. Perhaps the authors can indicate the significance of these data in relation to the hypothesis/scope of the study.

---

## Round 0.2 · Major Revisions

Thank you for making the changes, however there are still some areas which are not addressed well and need more improvement.
The fish (affected and unaffected) were collected from the same pond. Is it possible that the unaffected fish were possibly infected but still asymptomatic? How did you define healthy fish? How did you define EUS positive fish? Would it be possible that some of the fish used as healthy controls had microscopic lesions which could not be seen with naked eye? As the focus of the paper is comparison between those sick and healthy it is really important to have the right individuals.
I am most concerned about the histology results, Figure 1b is showing most likely dermis of EUS fish and skeletal muscle of unaffacted (labelled "healthy") fish, as these are different types of tissues they cannot be compared. In the text, damage to epithelia of the ulcerated skin is mentioned citing Fig 1B but this is not shown in Fig 1B. Please make sure that the affected and control histology can be compared, they should both show skin. It would be really great to include the edge of the ulcer (transition from normal to lesion). Please make sure that the figures are illustrating the text, particularly when you refer to the figures to support your statements .

I agree with the comments from the reviewers that the remarks about pathogenicity of some of the bacteria that DNA was detected is speculative and should be removed from Discussion. It should be deleted from conclusions, in particular the last sentence in conclusions is unsupported by the results and should be removed. While the microbiomes are different, it is unclear if the bacteria are really pathogenic as pathogenicity testing was outside the scope of this paper. Please do not overstate your results in Discussion and definitely not in Conclusions.

There are some discrepancies in your reply to reviewers comments. For example you state in the response that no bacteria were isolated, but in Line 111 you stated that samples were transferred to the laboratory for bacterial culturing. Were the samples sterile or were the culturing methods inappropriate? I cannot find the description of the bacteria culture methods.

Line 99 - should it be "naturally infected" instead of "spontaneously infected"?

Line 137 - should be "to confirm the presence of Aphanomyces" not to "identify the fungal pathogen"

---

## Round 0.3 · Minor Revisions

Thank you for revising the manuscript.
I have a few minor corrections:

As the fish collected as controls were not checked for all possible pathogens and diseases please change from:
"Healthy controls with normally swimming and feeding behavior were also collected from the same pond."
to
"Fish showing normal swimming and feeding behavior were also collected from the same pond."

Please change the description of the histology from:
"Muscle underlying the damaged skin of hybrid snakehead with EUS becomes ulcerous and is subsequently parasitized by numerous filamentous fungi (Fig. 1A). H&E-stained paraffin tissue sections showed damage to the muscle cells of the ulcerated muscle of hybrid snakehead with EUS (Fig. 1B). Numerous mycotic granulomas and large amounts of mycelia, which are typical characteristics of EUS, were detected in the ulcerated muscle of hybrid snakehead with EUS"
to
"Muscle underlying the damaged skin of hybrid snakehead with EUS became ulcerated and were colonised by numerous filamentous fungi (Fig. 1A, 1B). Control muscle shown for comparison (Fig. 1B). Numerous mycotic granulomas and large amounts of mycelia, which are typical characteristics of EUS, were detected in the ulcerated muscle of hybrid snakehead with EUS (Fig 1B)."

---

## Round 0.4 · accepted · Accept

Thank you for making all required changes.

#